# Study protocol for an international, multicentre stepped-wedge cluster randomised trial to evaluate the impact of a digital antimicrobial stewardship smartphone application

R I Helou ![ORCID],[1] Gaud Catho,[2] Annabel Peyravi Latif,[3] Johan Mouton,[1] M Hulscher,[4] Steven Teerenstra,[5] John Conly,[6] Benedikt D Huttner ![ORCID],[2] Thomas Tängdén,[3] Annelies Verbon[1]

For numbered affiliations see end of article.

**Correspondence to**
Professor Annelies Verbon;
a.verbon@erasmusmc.nl

## ABSTRACT

**Introduction** With the widespread use of electronic health records and handheld electronic devices in hospitals, informatics-based antimicrobial stewardship interventions hold great promise as tools to promote appropriate antimicrobial drug prescribing. However, more research is needed to evaluate their optimal design and impact on quantity and quality of antimicrobial prescribing.

**Methods and analysis** Use of smartphone-based digital stewardship applications (apps) with local guideline directed empirical antimicrobial use by physicians will be compared with antimicrobial prescription as per usual as primary outcome in three hospitals in the Netherlands, Sweden and Switzerland. Secondary outcomes will include antimicrobial use metrics, clinical and process outcomes. A multicentre stepped-wedge cluster randomised trial will randomise entities defined as wards or specialty regarding time of introduction of the intervention. We will include 36 hospital entities with seven measurement periods in which the primary outcome will be measured in 15 participating patients per time period per cluster. At participating wards, patients of at least 18 years of age using antimicrobials will be included. After a baseline period of 2-week measurements, six periods of 4 weeks will follow in which the intervention is introduced in 6 wards (in three hospitals) until all 36 wards have implemented the intervention. Thereafter, we allow use of the app by everyone, and evaluate the sustainability of the app use 6 months later.

**Ethics and dissemination** This protocol has been approved by the institutional review board of each participating centre. Results will be disseminated via media, to healthcare professionals via professional training and meetings and to researchers via conferences and publications.

**Trial registration number** ClinicalTrials.gov registry (NCT03793946). Stage; pre-results.

## INTRODUCTION

Antimicrobial drugs are an indispensable part of modern medicine, which depends on

---

### Strengths and limitations of this study

► This study will be an international randomised, multicentre, clinical trial using a rigorous design and methodology to evaluate the impact of an antimicrobial stewardship smartphone application (app) for the hospital setting.

► The app can easily be adapted to local guidelines which is a key feature for potential future use in other settings.

► The primary outcome is appropriate empiric antimicrobial therapy. The study is conducted in three hospitals with relatively high rates of appropriate antibiotic use, which may theoretically lead to a higher risk of a negative trial.

► In this cluster randomised trial, wards/specialties are the unit of randomisation and there is a risk of contamination through physicians changing between wards or among specialties, although this risk is lower than for a single-centre randomised trial. Restricted access to the software and tracking of app use will be applied to monitor possible contamination.

---

effective drugs for prophylaxis and treatment. Yet, only around 40%–70% of empiric antimicrobial regimens are considered appropriate with regard to their indication, the choice of agent, dosing or duration.[1–4] Inappropriate antimicrobial use enhances the risks of treatment failure and side effects in the individual patient, and accelerates selection and transmission of antimicrobial-resistant pathogens in healthcare settings and the community.[5] Many antimicrobial stewardship (AMS) interventions have been shown to increase appropriateness of antimicrobial drug prescribing.[6]

A software application (app), which is compatible with mobile electronic devices such as smartphones and tablets, and

increases the availability of existing national and local antibiotic guidelines plus local antibiograms, may improve antimicrobial prescribing, reduce medication errors and ultimately improve patient outcome. Due to its highly customisable nature, a digital app has the potential to become an effective AMS tool in all countries.

Mobile-software apps seem particularly interesting for the purpose of AMS, since they do not require access to electronic health records (EHRs) or computerised physician order entry systems, that are costly, difficult to maintain and not easily customisable. Most healthcare workers own a smartphone and often use it to access information to assist in their treatment decisions.[7 8] The vast majority of available apps developed to support prescription of medicines are limited to knowledge content and do not provide decision support algorithms. Further, they have rarely been evaluated in clinical trials.[9] To date, most reported digital AMS interventions are single-centre studies with limited internal validity due to methodological limitations and uncertain external validity.[4 10 11]

In the present study, we will evaluate the implementation and impact of a smartphone-based software app (the AB-Assistant, Spectrum, Calgary, Canada) adapted to local conditions in an international clinical trial using a stepped-wedge cluster randomised design and appropriateness of empiric antimicrobial prescribing as the primary outcome.

## METHODS AND ANALYSIS
### Setting
This study will be conducted in three academic, tertiary care hospitals in Europe: Erasmus University Medical Center, Rotterdam, the Netherlands; Uppsala University Hospital, Uppsala, Sweden and Geneva University Hospitals, Geneva, Switzerland (table 1).

### Study design
We plan to conduct an international, multicentre, stepped-wedge cluster randomised trial to assess whether

| Table 1 | Characteristics of the participating centres | | |
|---|---|---|---|
| **Name** | **Geneva University Hospitals** | **Erasmus MC Rotterdam** | **Uppsala University Hospital** |
| Website | www.hug-ge.ch/en/ | https://www.erasmusmc.nl/ | https://www.akademiska.se/en/ |
| Abbreviation | HUG | EMC | UUH |
| City | Geneva | Rotterdam | Uppsala |
| Country | Switzerland | The Netherlands | Sweden |
| Care level | Primary and tertiary care | Tertiary care | Tertiary care |
| Academic affiliation | Yes | Yes | Yes |
| Number of beds | About 1 900 | 900 | About 1 000 |
| Availability of antibiotic guidelines | Paper format and PDF<br><br>Updated every 2 years<br><br>Integrated into the EHR in some units in the context of a different study (Trial registration number: NCT03120975); COMPASS units not participating in this study | Website containing local guidelines. Updated every 2 years. Guidelines are based on the national guidelines of SWAB | Local guidelines in paper format and online (pdf). Updated every 2–3 years.<br><br>National guidelines provided by STRAMA available online and in app format, not routinely used at UUH. |
| Current standard of care antimicrobial stewardship | ▶ On demand ID consultations<br><br>▶ Review of all positive blood cultures<br><br>▶ Daily rounds on some units<br><br>▶ Approval required for certain antibiotics | ▶ On demand ID consultations<br><br>▶ Daily rounds on Intensive care units, weekly rounds on all units<br><br>▶ Review and feedback for certain antibiotics<br><br>▶ Approval required for certain antibiotics<br><br>▶ Review of all positive blood cultures | ▶ On demand ID consultations<br><br>▶ Daily rounds on intensive care units<br><br>▶ Adapted information and feedback to physicians at major departments 1–2 times per year |

EHR, electronic health record; ID, infectious disease; STRAMA, Swedish strategic programme against antibiotic resistance; SWAB, Dutch working party on antibiotic policy.

| Table 2 | PICOT of the study question |
|---|---|
| Population | Physicians involved in antimicrobial prescribing decisions for hospitalised adult patients in the participating centres |
| Intervention | Making a smartphone application with antimicrobial treatment recommendations available to physicians mentioned above |
| Comparator | Standard-of-care antimicrobial stewardship |
| Outcome | Appropriateness of empiric antimicrobial prescribing based on predefined criteria |
| Time | 12 months consisting of a 6 months introduction period with 6 months follow-up to assess sustainability |

making a smartphone app for AMS (Spectrum) can improve the appropriateness of antimicrobial prescribing in hospitalised patients. Table 2 illustrates the study question according to the Population, Intervention, Comparator, Outcome and Time (PICOT) framework.

In total, 36 medical and surgical entities (defined as wards (Sweden) or specialties (Switzerland, The Netherlands) will be included in the study using a stepwise design with six entities (two entities per participating hospital) 4-weekly at six introduction moments (figure 1). The primary outcome is appropriateness of empiric antimicrobial prescriptions assessed prior to implementation and up to 12 months after the intervention (after 6 months free use of the app). Empiric therapy is defined as[1] treatment started based on clinical evaluation before culture results are known,[2] the duration of treatment is longer than 24 hours (from the moment of prescription) and[3] the treatment is not specifically recorded as prophylactic. Initial targeted therapy for which there is a local guideline available (eg, *Clostridium difficile*) may be assessed for appropriateness.

The entities will be randomised as to the time period of making the app available (see the Intervention section). The order of implementation will be determined by a computer-generated list of random numbers programmed by a statistician otherwise not involved in the trial and AMS activities. Randomisation will be stratified by the type of specialty (medicine, surgery, etc). After a baseline period of 2-week measurements, six periods of 4 weeks will follow in which the intervention is introduced in six entities (in three hospitals) until all entities have implemented the intervention. We plan to include 36 entities in total (see sample size calculation below) during the six introduction moments and at the end of the inclusion time we will allow the use of the app by everyone, also wards/specialties not included in the study. In this way, the trend in appropriate antimicrobial use can be monitored per ward/specialty and the decrease of antimicrobial use and appropriateness of empirical antimicrobial therapy can be followed during the stepwise implementation.[12] At 12 months (after 6 months free use of the app) we will have a 2-week measurement period to evaluate the sustainability of the intervention. The study will start between February 2020 and June 2020, and end 12 months later.

## Justification of the stepped-wedge cluster randomised study design

The stepped-wedge cluster randomised design was chosen after considering several advantages and disadvantages of randomisation at the patient, physician, specialty and ward level. Patients are the 'classic unit' of randomisation and outcomes are often assessed at the patient level. However, there is a large risk of contamination of the intervention since one patient is often taken care of by

| Baseline No intervention | 0-4 weeks | 4-8 weeks | 8-12 weeks | 12-16 weeks | 16-20 weeks | 20-24 weeks | 6-12 months All wards using app |
|---|---|---|---|---|---|---|---|
| All hospital wards | | | | | | | ■ |
| Ward 31-36 | | | | | | ■ | ■ |
| Ward 25-30 | | | | | ■ | ■ | ■ |
| Ward 19-24 | | | | ■ | ■ | ■ | ■ |
| Ward 13-18 | | | ■ | ■ | ■ | ■ | ■ |
| Ward 7-12 | | ■ | ■ | ■ | ■ | ■ | ■ |
| Ward 1-6 | ■ | ■ | ■ | ■ | ■ | ■ | ■ |

**Figure 1** Study design. In all time periods (total of 7) there is an uptake period of 2 days and a measure period of 26 days. During the baseline period only a measure period of 2 weeks will be performed.

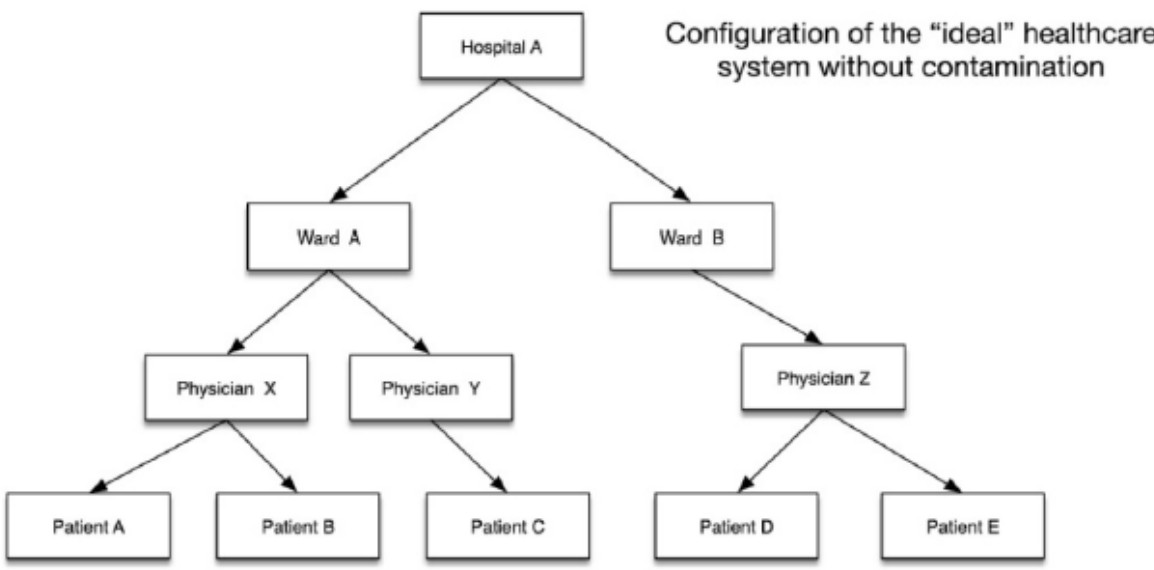

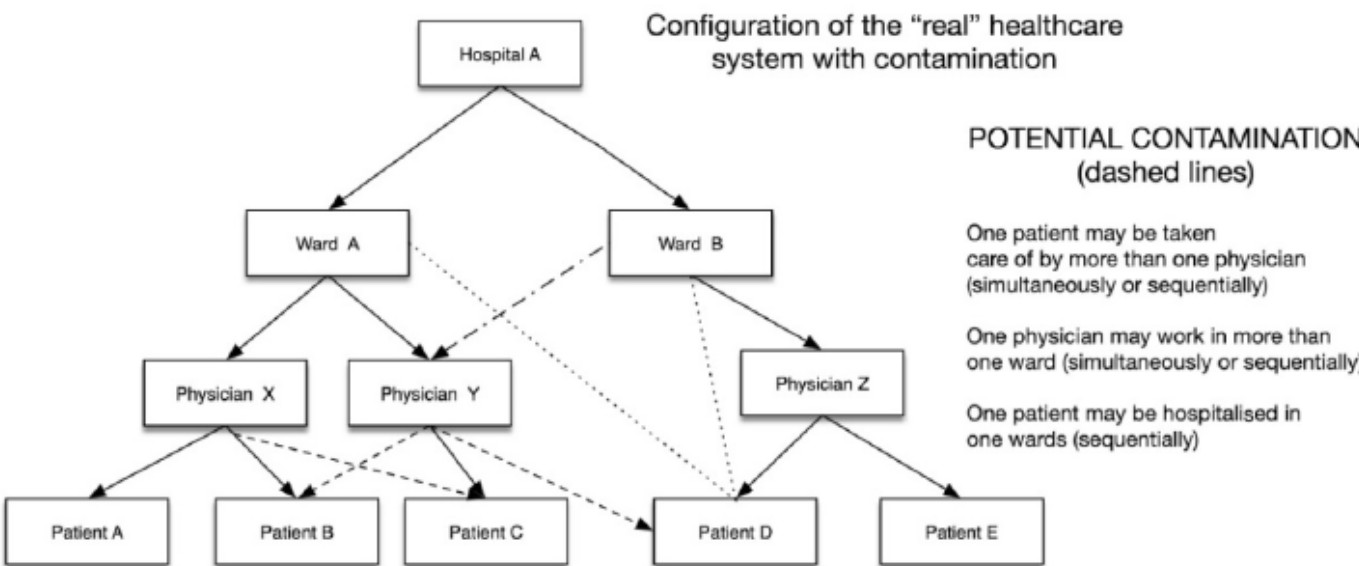

**Figure 2** Scheme of potential contamination. The potential contamination in a hospital, in and between wards, physicians and patients

many different physicians (see figure 2). Furthermore, the study intervention targets physicians and not patients, therefore precluding randomisation at the patient level as an option. Randomisation at the physician-level is, however, equally problematic since prescribing decisions in hospitals are often made by teams of physicians, making it difficult to assign a decision to a specific individual physician and introducing the risk of contamination between physicians within a team. We therefore decided to use the wards/specialty as entity of randomisation since it minimises the risk of contamination and reflects 'real-life' AMS in hospitals where interventions are usually implemented at a ward/specialty level.

The stepped-wedge design, which involves the randomised and sequential rollout of the intervention to clusters over time, was chosen because it can model and adjust for underlying temporal trends and results in all clusters having access to the intervention at the end of the rollout period which seems desirable from an AMS perspective.[13]

### Study population
In our study, the intervention will be targeted at physicians that prescribe antimicrobials. To avoid possible contamination, the prescribers will be randomised at the ward/specialty level and not individually (see above). Process parameters will be determined in physicians who have been randomised to use the app as per the process indicated above. Outcome parameters such as appropriate antimicrobial use will be determined at the patient level

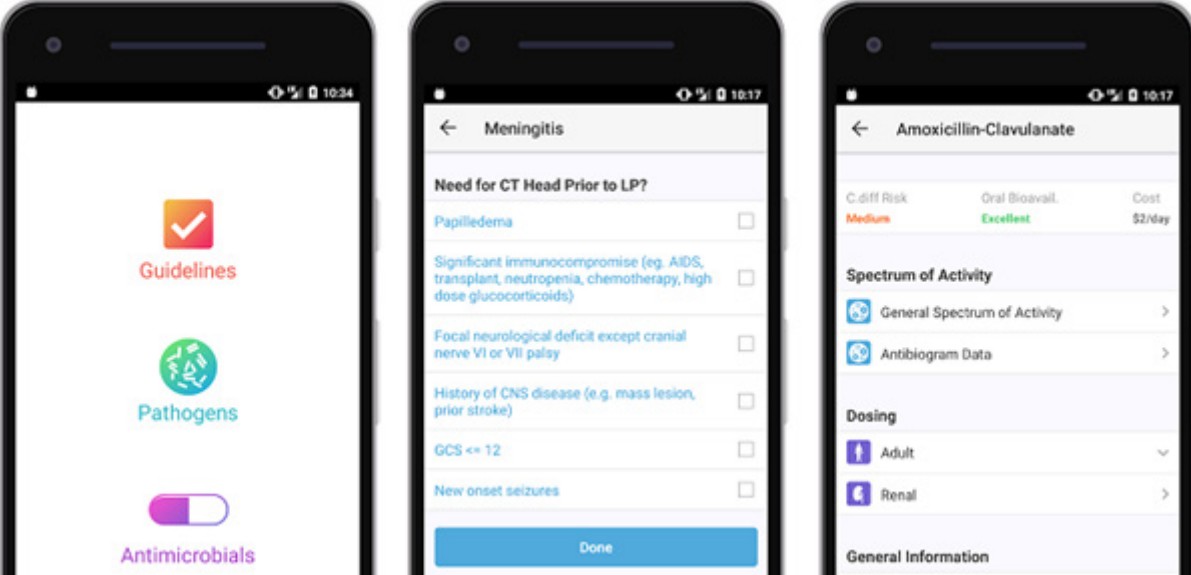

**Figure 3** Interface of the AB-assistant app. On the left the Home screen is shown. In the middle checkboxes are shown to guide the user to the correct therapy for Meningitis. The right screen shows details about Amoxicillin-Clavulanate regarding dosing and general spectrum of activity.

at participating wards (control and intervention) who meet the following criteria: being a medical or surgical specialty patient at least 18 years of age for whom antimicrobial drugs are prescribed.

### Intervention

The Spectrum app, available for iOS and Android and on the web, has been developed in Canada (www.spectrum.app) and is now being used in over 50 hospitals, mainly in Canada and the USA. The app has been developed with the help of human factor experts and integrates multiple resources into a unified user-friendly interface for use at point-of-care to improve antimicrobial prescribing (figure 3).

For the purpose of this study the app and the content management system were extended to also support languages other than English (French, Dutch and Swedish) and will be used for prescribing guidelines, pathogens, antimicrobial drugs and antibiograms. App content such as local guidelines, information regarding antimicrobial drugs, and micro-organisms of each hospital was entered by investigators of the Dutch, Swiss and Swedish hospitals (RH, GC, BH and APL). App content is identical to content of the local guidelines if these are published on a website. Minor discrepancies may occur if the local guidelines are published in a digital document (ie, PDF) or on paper. The app can be more up to date in those cases.

The allocation of the timing to introduce the app will be concealed to the participating cluster up until 2 weeks before initiating use of the intervention. At the start of the intervention physicians on the intervention ward(s) will be informed about the function and content of the app. During the trial, only physicians on the intervention wards will be whitelisted to be able to use the app. Access

to the app will be centrally withdrawn if a physician leaves the intervention ward (eg, to work on a different (ie, non-intervention) ward). The conventional route of guidelines (eg, portable document formats (PDF), website or printed booklet) will still be available at all times. Physicians can freely choose between the app and the conventional route. Physicians on control wards will only be able to access the guidelines using the conventional route.

### Patient and public involvement

No patients or public have been included in the study design.

## OUTCOME PARAMETERS

Outcome parameters (table 3) will be monitored using a digital case report form (CRF) (online supplementary appendix 1) in the software application OpenClinica.[14]

### Primary outcome parameter

Appropriateness of empiric antimicrobial therapy expressed as number and percentage of appropriate treatment episodes compared with all assessed treatment episodes.

Definitions for the primary outcome are:
- ► Antimicrobial drugs assessed are anatomical therapeutic chemical (ATC) classes J01, J02, J03, J04, J05 (excluding anti-HIV drugs), P01, P02 and P03.
- ► Treatment episodes are defined as the duration of treatment given for a specific infectious episode with gaps of treatment of no longer than 1 day.
- ► Treatments are antimicrobials started during the measurement period at the participating entity. If a treatment was started within 24 hours before transfer in the emergency room or other ward and has not been changed during the first 24 hours stay at the

**Table 3** Secondary study parameters/endpoints and other study parameters

| Outcome type | Examples | Data source |
|---|---|---|
| Quantitative antimicrobial use | ▶ Total prescription of antimicrobial drugs1 in DDD/admission, DDD/PD, DOT/PD and DOT/admission | ▶ EHR |
| | ▶ Total prescription of broad spectrum and restricted antimicrobial drugs in DDD/admission | ▶ Administrative data |
| | ▶ Total prescription of antimicrobial drugs per AWaRe category[19] in DDD/admission | |
| | ▶ Antimicrobial costs | |
| Patient | ▶ Length of hospital stay | ▶ EHR |
| Related | ▶ In hospital mortality within 30 days after admission (all cause) | ▶ Administrative data |
| | ▶ Unplanned hospital readmissions within 30 days after discharge | |
| | ▶ Transfer to intermediate care or ICU within 30 days after admission | |
| Microbiologic and HAI | ▶ Incidence of healthcare facility onset *Clostridium difficile* | ▶ EHR |
| | ▶ Incidence clinical cultures with multi-drug resistant organisms (MRSA, ESBL-E, CPE, VRE, multidrug resistant *Pseudomonas aeruginosa*) denominated per 1000 PD and admission | ▶ Microbiology database |
| | | ▶ Infection control surveillance data |
| Physician related | ▶ Uptake of the AB-assistant (total users and number of sessions per user, time spent per session, time spent per screen, number of times each screen is viewed) | ▶ Content Management System |
| | ▶ Differences in uptake between centres | ▶ Content Management System |
| | ▶ Actual use of app and experiences while using it | ▶ Survey |
| Other outcomes | ▶ Number of infectious diseases consultations | ▶ EHR |

1. Antimicrobials belonging to Anatomical Therapeutic Chemical Classification System class J01, J02, J03, J04, J05 (excluding anti-HIV drugs), P01, P02 and P03, oral vancomycin (A07AA09) and fidaxomicin (A07AA12).
AWaRe, access, watch, reserve; CMS, content management system; CPE, carbapenemase-producing enterobacteriaceae; CRE, carbapenem-resistant enterobacteriaceae; DDD, defined daily dose ; DOT, days of therapy; EHR, electronic health record; ESBL-E, extended-spectrum beta-lactamase-producing E; HAI, hospital acquired infections; ICU, intensive care unit; MRSA, methicillin resistant staphylococcus aureus; PD, patient-days.

ward, the treatment will be considered in accordance with the choice of the ward physician.
▶ Treatments are in patients either newly admitted to the participating entities or already hospitalised during the preintervention period.
▶ Appropriateness of empiric therapy includes choice, route and the dose of the antimicrobial drugs.
▶ Appropriateness of therapy is in concordance with local treatment guidelines. If no guidelines are available, or if the treatment deviates from the guidelines or if treatment is guideline concordant but treatment should have deviated from the guideline (eg, because of allergies, colonisation with multidrug-resistant bacteria, other microbiological results, interactions, comorbidities or recent exposure to antimicrobials), anonymised cases will be discussed between two independent local infectious disease (ID) physicians. If they do not agree a third independent local ID physician will be consulted to adjudicate.
▶ For each patient only the first treatment episode during the measurement period will be evaluated. For patients treated for >1 infection (eg, pneumonia and *C. difficile* infection) appropriateness will be assessed for all infections combined (eg, if treatment is appropriate for *C. difficile* infection but inappropriate for

pneumonia the treatment would be categorised as inappropriate).

### Secondary outcome parameters
▶ Total prescription of antimicrobial drugs on the cluster-level expressed as defined daily dose (DDD) and day of therapy (DOT) denominated by patient-days and admission; subcategories by type of antimicrobial (antibiotic, antifungal, etc), WHO Access, Watch, Reserve (AWaRe) categories and restricted antimicrobial drugs (table 3).
▶ Clinical secondary outcomes: length of hospital stay, in hospital mortality and readmissions within 30 days after discharge, incidence/prevalence of key antimicrobial resistant (AMR) pathogens such as *C. difficile*.
▶ Process parameters of the intervention: penetration of the AB-assistant (total users and number of use of app), and uptake differences between centres.

Along side the randomised controlled trial (RCT), an uptake analysis will be performed to assess actual use of the app as well as an evaluation of use.

### Uptake analysis
A number of parameters regarding use of the app will be collected and made available on a secure dashboard

to authorised users at each site. These include number of monthly users and monthly sessions, number of times each antimicrobial, pathogen or clinical pathway screen is viewed, use by hospital, department and medical specialism. These data will enable in-app behaviour to be correlated with actual prescribing behaviour on ward level and site-wide basis, thereby strengthening the association of the intervention to observed outcomes.

## EVALUATION OF USE

Insufficient user friendliness and lack of integration into the workflow are key impediments to the uptake of IT-based AMS interventions. Therefore, we will evaluate the physicians' experiences with the use of the app. After each introduction period all physicians of intervention wards will receive a questionnaire to collect information on experiences using the app. The questionnaire will be based on information from the literature,[15] see online supplementary appendix 2.

## BLINDING DURING EVALUATION OF OUTCOME PARAMETERS

Given the nature of the intervention and the primary outcome assessed it will be difficult to blind the outcome assessor. To assure that assessment bias is limited we will have a subset of outcomes (10%) assessed by an independent investigator blinded to the study period.

## STATISTICAL METHODS

Baseline characteristics of patients (eg, gender, age, allergies and comorbidities), use of antimicrobial drugs, appropriate antimicrobial therapy and other outcome parameters will be described by entity.

To mitigate possible differences in time trends and baseline characteristics at patient and centre level, primary analysis will be by centre and effect estimates will be combined over centres using meta-analytic methods. These primary analyses will employ multivariable multilevel logistic models with time periods as fixed effects to adjust for time trend and treatment as fixed effect to estimate the OR of treatment. Entities (ward/specialty) will be included as a random effect to account for the correlation of patients within entities, assuming this correlation is independent of time period of including the patient. Moreover, covariates will be included to adjust for age, gender, hospital and specialty (medical, surgical). Similar models but using (generalised) linear multilevel models will be used for continuous outcomes or incidence/rate outcomes.

Sensitivity analyses may include the following: adjusting for additional confounders if imbalance at baseline is present; investigating deviations from the assumption that the correlation of patients within entities is independent of time period of including the patient; estimating differences in percentage (instead of odds ratios) using a (generalised) linear multilevel models with identity link and binomial error distribution and incorporating the three centres in one model.

Subgroup analysis for specialty and other factors as appropriate will be conducted.

## Sample size

By including 24 entities (wards/specialties) with seven measurement periods (including one baseline period) in which the primary outcome (appropriate empiric antimicrobial drug use) will be measured in 15 participating patients per entity per time period and an assumed intra-entity correlation coefficient of 0.1, the study has a power of 81% to detect an absolute improvement in appropriate antimicrobial use by 10 percentage points from of 60% to 70%.[16] To account for potential drop-outs or a lower inclusion rate, we aim to include 36 clusters in the three participating hospitals.

## ETHICS AND DISSEMINATION

This protocol has been approved by the institutional review board (IRB) of each participating centre. Ethical approval has been obtained from the Medical Ethics Committee Erasmus MC, Rotterdam, the Netherlands, number MEC-2019–0172, the Swedish Ethical Review Authority, Uppsala, Sweden, number 2019-05075 and from the Commission cantonaled'éthique de la recherche, Geneva, Switzerland, number 2019-01061. Physicians will be informed about the app without obligation to use it, furthermore analysis of usage will be anonymous, therefore a waiver for informed consent for physicians has been asked. However, participating physicians will be asked for a digital informed consent the first time they use the app. To monitor patient data, consent will be asked according to the country legislation and IRB mandate.

## Dissemination

Results will be disseminated to healthcare professionals and researchers via presentations at national and international conferences and publications in peer-reviewed journals. Furthermore, we plan to have a workshop with interested healthcare workers after the end of the study to disseminate experiences and findings. The public will be informed through press releases.

General project data and metadata will be included in the supplementary data of published articles if needed, and/or shared through university repositories.

## DISCUSSION AND IMPLICATIONS

To date, few IT-based AMS interventions have been evaluated in randomised clinical trials.[4 10 11] Clinical decision making by means of clinical decision support systems (CDSS) integrated in the EHR present the 'gold standard' of IT-based AMS. However, their widespread use has been limited by some major obstacles: (1) EHRs are still not universally implemented in all hospitals in high-income countries, and even less so in low-income and

middle-income countries (2) even if an EHR is implemented it usually lacks essential components for CDSS such as a computerised physician order entry system and (3) EHRs—especially those of commercial vendors—have designs that are difficult to modify/adapt. By developing an 'app' as a software programme running on smartphones and other mobile devices such as tablets, an intermediate or complementary step to full CDSS may be offered. Spectrum is a smartphone-based digital stewardship app that is customisable to local guidelines by local AMS teams. In our stepped-wedge cluster randomised trial we will evaluate the impact of this app on appropriate antimicrobial prescribing in high-income countries. However, smartphone adoption is consistently strong in middle-income countries such as in the Caribbean and reached low-income countries in Africa.[17 18] More and more healthcare workers have their own electronic devices (smartphone or tablets).[7] AMS smartphone-based interventions can be made easily available, in high-income, low-income and middle-income countries. The ability of this app will make antibiotic prescription guidelines more accessible for physicians all over the world, thereby helping to fight AMR.

**Author affiliations**
[1]Department of Medical Microbiology and Infectious Diseases, Erasmus Medical Center, Rotterdam, The Netherlands
[2]Department of Infectious Diseases, Hopitaux Universitaires de Geneve, Geneva, Switzerland
[3]Department of Infectious Diseases, Uppsala University Hospital, Uppsala, Sweden
[4]Scientific Center for Quality of Healthcare (IQ Healthcare), Radboud Institute for Health Sciences, Radboud University Medical Center, Nijmegen, The Netherlands
[5]Department for Health Evidence, Radboud Institute for Health Sciences, Radboud University Medical Center, Nijmegen, The Netherlands
[6]Department of Medicine, University of Calgary and Alberta Health Services, Calgary, Alberta, Canada

**Contributors** A draft protocol was written by AV, BDH, TT, JC and JM. During extensive calls and meetings with AV, BDH, TT, JC, JM, RIH, GC, APL, MH and ST the introduction, methods and analysis, outcome parameters, evaluation of use, blinding during evaluation of outcome parameters, statistical methods, ethics and dissemination and discussion and implications were further shaped and fine-tuned.

**Funding** This work is supported by Joint Programming Initiative on Antimicrobial Resistance (JPIAMR), grant number JPIAMR2017-045.

**Competing interests** None declared.

**Patient and public involvement** Patients and/or the public were not involved in the design, or conduct, or reporting or dissemination plans of this research.

**Patient consent for publication** Not required.

**Provenance and peer review** Not commissioned; externally peer reviewed.

**ORCID iDs**
R I Helou http://orcid.org/0000-0002-0448-6349
Benedikt D Huttner http://orcid.org/0000-0002-1749-9464

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
