## [Reviewer comments · BMJ Open]

ARTICLE DETAILS

TITLE (PROVISIONAL)	Study protocol for an international, multicenter stepped-wedge cluster randomized trial to evaluate the impact of a digital antimicrobial stewardship smartphone application
AUTHORS	Helou, R.I.; Catho, Gaud; Peyravi Latif, Annabel; Mouton, Johan; Hulscher, M; Teerenstra, Steven; Conly, John; Huttner, Benedikt D; Tängdén, Thomas; Verbon, Annelies

VERSION 1 - REVIEW

REVIEWER	David Andresen St Vincent's Hospital, Sydney, Australia
REVIEW RETURNED	14-Oct-2019

GENERAL COMMENTS	The protocol is well-written and the design is appropriate for the question being asked. I do not have any direct experience with the statistical analysis or a stepped-wedge trial but assume that the authors have taken appropriate advice. I think that the power calculation does not really belong in the abstract. I am a bit confused about the analysis of 15 patients per entity - does that mean that only the first 15 eligible patients per entity per time period will be analysed? If so please state this explicitly. Please also define empiric (as opposed to targeted and prophylactic) antibiotic prescription, and how this will be adjudicated. Am I right in assuming that only the first 15 EMPIRIC patients will be included and that all targeted and prophylactic AB's are essentially ignored? What about empiric use that continues and "becomes" targeted once microbiology results are available? Please elaborate on the blind assessment of 10% of outcomes, and how these results will be handled? For instance, what proportion of discrepant results would trigger blinded review of all patients? Nice work and I look forward to reading the results!
---

REVIEWER	M van Agtmael and M Bakkum Amsterdam UMC, Internal Medicine Netherlands
REVIEW RETURNED	30-Oct-2019

GENERAL COMMENTS	Dear authors,
---------------

	I would like to congratulate you with this well-written protocol, aiming to investigate a promising tool in reducing antimicrobial resistance. I believe the study is well-designed and the choice for the design is very clearly justified in the according paragraph. I just have some minor inquiries that did not become completely clear to me. I believe providing an answer to my questions will benefit the protocol.  - Besides your global time-line showing the inclusions per four week period, I believe it to be helpful if you report the actual dates in which you aim to start this study (or have already started?). - The Spectrum app has more functionalities than just the guideline and formulary. I believe it to be insufficiently clear from the protocol whether you aim to use all of these functionalities, including the antibiogram results and infection-prevention guidelines or whether you stick to the prescribing guidelines and perhaps the formulary. - You mention as one of the study limitations that all western European centers are likely to have already high guideline-compliance rates and low rates of antimicrobial resistance. Moreover, you show in table 1 that the current availability of antibiotic guidelines varies significantly between the centers. For example the mobile accessible websites of the Dutch SWAB system may not actually be that much different from the app, whereas the paper and PDF guidelines in the other centers are. Besides statistically correcting for this difference, it may be interesting to perform an in-depth analysis to any differences between the centers that you may find (are the Dutch less likely to use the app, because the SWAB system works just fine?). Adding some qualitative, open questions, may help you to do so? (and might give some other helpful insights as well). - Page 7, intervention subheading: You have taken great care to ensure that the information shown in the app is correct. However, nowhere do I read that this information is exactly the same as in the current guidelines. As you aim to investigate the medium (app) rather than new guidelines, I believe it to be very important that you verify that they are in fact the same (or explain why they are not). - Page 8, bullet point on line 26: It is great that you have a protocol on how to cope with guideline deviations. But such deviations can, by definition, not be provided in the guidelines or the app. Therefore any differences that you may find will remain unexplained. Moreover, as you do not mention that the infectious disease specialists are blinded while discussing the case, it opens up a possibility for bias. Perhaps you can consider blinding the specialists or perhaps it is better to exclude all cases that do not fit the guideline? - Page 10, line 49: "Physicians will be informed...". The part of this sentence where you explain that a waiver for consent has been asked is confusing, as it appears that this waiver was just asked but not (yet) provided. Whether the need for informed consent is waived has important consequences for the feasibility of the study (it might be very difficult for you to get consent of all participating doctors). Therefore, I believe it is important to clarify this sentence.
--	---

VERSION 1 – AUTHOR RESPONSE

Reviewer: 1

Reviewer Name: David Andresen

Institution and Country: St Vincent's Hospital, Sydney, Australia Please state any competing interests or state 'None declared': None

1. I think that the power calculation does not really belong in the abstract. I am a bit confused about the analysis of 15 patients per entity - does that mean that only the first 15 eligible patients per entity per time period will be analysed? If so please state this explicitly.

Thank you for this comment. The power calculation has been removed from the abstract.

We will try to assess a random sample of 15 participating patients per entity per time period. We have clarified this in the revised manuscript on page 9, line 289.

2. Please also define empiric (as opposed to targeted and prophylactic) antibiotic prescription, and how this will be adjudicated.

Empiric therapy is defined as (1) treatment started based on clinical evaluation before culture results are known, (2) the duration of treatment is longer than 24 hours (from the moment of prescription) and (3) the treatment is NOT specifically recorded as prophylactic.

This has been clarified in the manuscript on page 4, lines 121 to 125.

3. Am I right in assuming that only the first 15 EMPIRIC patients will be included and that all targeted and prophylactic AB's are essentially ignored?

Patients treated according to local empiric treatment guidelines will be predominantly included, however some patients might be initially treated according to a local targeted guideline (e.g. C. difficile) and may be included. Also see the answers in question 1 & 2.

4. What about empiric use that continues and "becomes" targeted once microbiology results are available?

Initial empiric therapy will be assessed for appropriateness according to the local guidelines as well as time to streamline the therapy (empiric \square targeted) from the moment microbiological test results are known. The subsequent targeted therapy will not be assessed for appropriateness. Initial targeted therapy for which there is a local guideline available may be assessed for appropriateness.

5. Please elaborate on the blind assessment of 10% of outcomes, and how these results will be handled? For instance, what proportion of discrepant results would trigger blinded review of all patients?

A random sample of 10% of all included patients will be assessed by an independent investigator blinded to the study period. A discrepancy greater than 20% regarding appropriate prescribed antimicrobial treatment of the sample would trigger a review of 30% of all included patients.

Nice work and I look forward to reading the results!

Thank you very much.

Reviewer: 2

Reviewer Name: M van Agtmael and M Bakkum Institution and Country:

Amsterdam UMC, Internal Medicine

Netherlands

Please state any competing interests or state 'None declared': none

1. Besides your global time-line showing the inclusions per four week period, I believe it to be helpful if you report the actual dates in which you aim to start this study (or have already started?).

Thank you for this comment. Our aim is to start the study between January 1, 2020 and April 1, 2020. This does not change the content of the manuscript and we have chosen not to include the actual start dates.

2. The Spectrum app has more functionalities than just the guideline and formulary. I believe it to be insufficiently clear from the protocol whether you aim to use all of these functionalities, including the antibiogram results and infection-prevention guidelines or whether you stick to the prescribing guidelines and perhaps the formulary.

Thank you for bringing up this point. We will use the Spectrum app for prescribing guidelines, pathogens, antimicrobial drugs and antibiograms. We have clarified this in the manuscript on page 6, lines 176 to 177.

3. You mention as one of the study limitations that all western European centers are likely to have already high guideline-compliance rates and low rates of antimicrobial resistance. Moreover, you show in table 1 that the current availability of antibiotic guidelines varies significantly between the centers. For example the mobile accessible websites of the Dutch SWAB system may not actually be that much different from the app, whereas the paper and PDF guidelines in the other centers are. Besides statistically correcting for this difference, it may be interesting to perform an in-depth analysis to any differences between the centers that you may find (are the Dutch less likely to use the app, because the SWAB system works just fine?). Adding some qualitative, open questions, may help you to do so? (and might give some other helpful insights as well).

Thank you for these suggestions. The content of the app guidelines will be identical with the current guidelines, whether these are on a website or on paper. We indeed plan to analyse differences between countries in uptake of the app, however, this has not been defined as an endpoint and we, therefore, do not describe this in the manuscript. A questionnaire will be distributed as described in the manuscript on page 8, lines 263 to 264.

4. Page 7, intervention subheading: You have taken great care to ensure that the information shown in the app is correct. However, nowhere do I read that this information is exactly the same as in the current guidelines. As you aim to investigate the medium (app) rather than new guidelines, I believe it to be very important that you verify that they are in fact the same (or explain why they are not).

The guidelines in the app are indeed the same as the current guidelines if displayed on a website of the hospital. However, if the current guidelines are in a digital document (i.e. PDF) or on paper the app content can be more up to date. This is one of the advantages of the app: the guidelines can be updated in real time. This has been added in the manuscript on page 6, lines 179 to 182.

5. Page 8, bullet point on line 26: It is great that you have a protocol on how to cope with guideline deviations. But such deviations can, by definition, not be provided in the guidelines or the app. Therefore any differences that you may find will remain unexplained. Moreover, as you do not mention that the infectious disease specialists are blinded while discussing the case, it opens up a possibility for bias. Perhaps you can consider blinding the specialists or perhaps it is better to exclude all cases that do not fit the guideline?

Thank you. The infectious disease specialists discussing the case will not be the ones involved in the actual patient care and will be blinded to the study period.

6. Page 10, line 49: "Physicians will be informed...". The part of this sentence where you explain that a waiver for consent has been asked is confusing, as it appears that this waiver was just asked but not (yet) provided. Whether the need for informed consent is waived has important consequences for the feasibility of the study (it might be very difficult for you to get consent of all participating doctors). Therefore, I believe it is important to clarify this sentence.

We have discussed this item with the IRBs in the participating centres. Participating physicians will be asked a digital informed consent and this has been added in the manuscript on page 10, lines 302 to 303.

VERSION 2 – REVIEW

REVIEWER	David Andresen St Vincent's Public Hospital, Sydney, Australia
REVIEW RETURNED	03-Jan-2020

GENERAL COMMENTS	All issues with the first version have been fully addressed. Good luck with the study!
--

REVIEWER	M van Agtmael and M Bakkum Amsterdam UMC, the Netherlands
REVIEW RETURNED	27-Jan-2020

GENERAL COMMENTS	Congratulations on this well-written protocol for a promising study. I have just some minor concerns regarding the rebuttal. 1. The author's have chosen not to include the dates of the study, the reasons for this are explained. However, the journal editors have specifically asked us to check for dates in the review process. If the dates are not entirely clear at this moment, perhaps it is a good idea
---

	to mention ranges in which the study is planned to start and end in (e.g. between january and april 2020). 2. The author's mention (rebuttal point 3) that they plan to look at the difference in uptake of the application between countries. If this is the case, we believe that it should be mentioned in the study protocol. Moreover, as we suggested previously, the 5-point Likert-type questions in the post-intervention survey are convenient for both the researchers and participants, but open-ended questions (w/ qualitative analysis) usually provide much more valuable information. This is an opportunity to add them to the post-intervention survey.
--	--

VERSION 2 – AUTHOR RESPONSE

Congratulations on this well-written protocol for a promising study. I have just some minor concerns regarding the rebuttal.

1. The author's have chosen not to include the dates of the study, the reasons for this are explained. However, the journal editors have specifically asked us to check for dates in the review process. If the dates are not entirely clear at this moment, perhaps it is a good idea to mention ranges in which the study is planned to start and end in (e.g. between january and april 2020).

2. The author's mention (rebuttal point 3) that they plan to look at the difference in uptake of the application between countries. If this is the case, we believe that it should be mentioned in the study protocol. Moreover, as we suggested previously, the 5-point Likert-type questions in the post-intervention survey are convenient for both the researchers and participants, but open-ended questions (w/ qualitative analysis) usually provide much more valuable information. This is an opportunity to add them to the post-intervention survey.